

# Formation mechanism and source apportionment of water-soluble organic carbon in PM$_1$, PM$_{2.5}$ and PM$_{10}$ in Beijing during haze episodes

Qing Yu[1,2], Jing Chen[1,2], Weihua Qin[1,2], Yuepeng Zhang[1,2], Siming Cheng[1,2], Mushtaq Ahmad[1,2],
Xingang Liu[1,2], Hezhong Tian[1,2]

[1]State Key Joint Laboratory of Environment Simulation and Pollution Control, School of Environment, Beijing Normal
University, Beijing 100875, China.
[2]Center of Atmospheric Environmental Studies, Beijing Normal University, Beijing 100875, China.

*Correspondence to*: Jing Chen (jingchen@bnu.edu.cn)

**Abstract.** Water soluble organic carbon (WSOC) in atmospheric aerosols may pose significant impacts on haze formation,
climate change, and human health. This study investigated the distribution characteristics and sources of WSOC in Beijing
based on the diurnal PM$_1$, PM$_{2.5}$ and PM$_{10}$ samples collected during haze episodes in winter and early spring of 2017. The
haze episode in winter showed elevated level of WSOC, around three times of that in spring. WSOC was enriched in PM$_{2.5}$
in winter while the proportions in both finer (0-1 μm) and coarse particles (2.5-10 μm) increased in spring. Several organic
tracers were carefully selected and measured to demonstrate the sources and formation mechanism of WSOC. Most of the
identified organic tracers showed similar seasonal variation, diurnal change and size distributions with WSOC, while the
biogenic secondary organic aerosol (SOA) tracer *cis*-pinonic acid was an obvious exception. Based on the distribution
characteristics of the secondary organic tracers and their correlation patterns with key influencing factors, the importance of
the gas-phase versus aqueous-phase oxidation processes on SOA formation was explored. The gas-phase photochemical
oxidation was weakened during haze episodes, whereas the aqueous-phase oxidation became the major pathway of SOA
formation, especially in winter, at night and for the coarser particles. Secondary sources accounted for more than 50% of
WSOC in both winter and spring. Biomass burning was not the dominant source of WSOC in Beijing during haze episodes.
Primary sources showed greater influence on finer particles while secondary sources became more important for coarser
particles during haze episode in winter. SOC estimated by the OC-EC method, WSOC-levoglucosan method, and PMF-
based methods were comparable, and the potential errors for different SOC estimation methods were discussed.



## 1 Introduction

Organic aerosols constitute a significant fraction (20-60 % by mass) of atmospheric particulate matter, in which water-soluble organic carbon (WSOC) accounts for 20-80 % of total organic carbon (Jaffrezo et al., 2005; Du et al., 2014; Tang et al., 2016). WSOC can alter the hygroscopicity of atmospheric aerosols, thus affecting aerosol size distribution and their ability to act as cloud condensation nuclei (CCN) (Ervens et al., 2011). Besides, WSOC may pose significant risk to human health as it is closely associated with the formation of reactive oxygen species (ROS) and cytotoxicity of atmospheric aerosols (Velali et al., 2016; Bae et al., 2017).

Due to the high proportion of WSOC in atmospheric aerosols and their environmental impact, the sources of WSOC have been widely discussed in the literature with a general recognition that WSOC mainly originates from direct emissions of biomass burning and secondary formation through the oxidation of volatile organic compounds in the atmosphere (Ding et al., 2008b; Feng et al., 2013; Du et al., 2014). Nevertheless, previous studies have also shown that primary emission sources other than biomass burning, such as vehicular exhaust emission, residual oil combustion, etc., also contribute to the WSOC load in the atmosphere (Kawamura and Kaplan, 1987; Guo et al., 2015; Kuang et al., 2015). Such primary emissions may even surpass biomass burning and become the major sources of WSOC (Kaul et al., 2014). Compared to the primary sources, estimate of the contributions of secondary components has been difficult when studying the sources of WSOC. For example, the chemical mass balance (CMB) model, a typical receptor model frequently used in aerosol source apportionment, can not quantify the specific contribution of each secondary source due to the lack of source profiles (Zheng et al., 2006; Ding et al., 2008a). The tracer-yield method based on the chamber experiments usually ignores the cloud process and the subsequent aqueous-phase reactions, thus may bring about large uncertainties when applying the secondary organic aerosol (SOA) yield results obtained under simple chamber conditions to the actual atmosphere (Kleindienst et al., 2007; Feng et al., 2013). In contrast, the positive matrix factorization (PMF) model combined with organic tracers has proved to be effective in quantifying the contributions of different secondary as well as primary sources of WSOC. For example, Feng et al. (2013) reported the sources and seasonal variations of WSOC and SOA in Shanghai based on PMF model and several SOA tracers.

The concentration, composition and sources of WSOC in atmospheric aerosols show significant regional and seasonal variations in China and worldwide (Jaffrezo et al., 2005; Ervens et al., 2011; Feng et al., 2013; Du et al., 2014; Kuang et al., 2015). As SOA takes a large proportion of WSOC, the sources of WSOC are in fact greatly impacted by meteorological conditions (Kaul et al., 2014). North China, particularly the Beijing-Tianjin-Hebei region, has been subject to frequent regional haze episodes in recent years. Research on haze formation and evolution showed that the highly polluted haze episodes were usually associated with high relative humidity and increased water-soluble fraction of $PM_{2.5}$ (Chen et al., 2014; Tian et al., 2014; Cheng et al., 2015). As water-soluble ions and the formation of secondary inorganic aerosols in haze episodes have been extensively studied, few studies focused on the water-soluble organic compounds and SOA formation. In fact, the formation of SOA during haze episodes can be influenced by the elevated levels of anthropogenic pollutants such as sulfate and $NO_x$ (Hoyle et al., 2011; Xu et al., 2015). Therefore, identifying the contributions of primary sources and





secondary formation to WSOC during haze episodes would help to elucidate the complex nature of WSOC and further put forward the effective control measures.

In this study, diurnal $PM_1$, $PM_{2.5}$ and $PM_{10}$ samples were collected at an urban site of Beijing during haze episodes in January, March, and April, 2017. The carbonaceous (OC, EC, WSOC) contents, water-soluble ions and typical organic tracers in each sample were measured. The contributions of primary versus secondary sources to WSOC during the sampling
periods were identified using PMF. The objectives of this study were to (1) clarify the distribution characteristics of WSOC in $PM_1$, $PM_{2.5}$ and $PM_{10}$ during haze episodes, (2) quantify the contributions of primary and secondary sources to WSOC in $PM_1$, $PM_{2.5}$ and $PM_{10}$, (3) and elucidate the formation mechanism of secondary components in WSOC during haze episodes. In addition, estimates of SOC via different methods were also compared and evaluated.

## 2 Experimental

### 2.1 Field sampling

The sampling site was located on the roof of the School of Environment Building (about 20 m above ground) in Beijing Normal University, representing a typical urban environment. Diurnal $PM_1$, $PM_{2.5}$ and $PM_{10}$ samples were collected in winter and early spring of 2017, including the winter period of Dec. 31, 2016-Jan. 10, 2017, and the spring periods of Mar. 15-Mar 25, 2017 and Apr. 3-Apr. 7, 2017, covering haze episodes during the sampling periods. The $PM_1$, $PM_{2.5}$ and $PM_{10}$
samples were simultaneously collected on pre-baked (500 °C for 4 h) quartz filters (PALLFLEX, 90 mm) using three independent medium-volume air samplers (Qingdao Hengyuan Technology Development Co., Ltd., HY-100) at a flow rate of 100 L min$^{-1}$. The daytime samples were collected from 8:00 to 19:30 and the nighttime samples from 20:00 to 7:30 the next day. After being stabilized under constant temperature (25 °C) and humidity (50 %), the filters were precisely weighed using an analytical balance (Sartorius, BSA124S, reading precision 0.1 mg) before and after sampling. The sample filters
were sealed in polyethylene bags and stored below -20 °C in a refrigerator for further analysis. In total, 63 and 90 samples were collected for the haze episodes in winter and spring respectively.

### 2.2 Chemical analysis

Seven organic tracers were measured for each sample, including levoglucosan, cholesterol, 4-methyl-5-nitrocatechol, phthalic acid, 2-methylerythritol, 3-hydroxyglutaric acid, and *cis*-pinonic acid. One-fourth of each sample filter was cut into
pieces and ultrasonically extracted with 10 mL methanol for 20 min for three times. The combined extracts were filtrated through a 0.45 μm PTFE syringe filter, concentrated using a rotary evaporator, and then blown to dryness under a gentle stream of ultrapure nitrogen. A mixture of 100 μL pyridine and 200 μL N,O-bis-(trimethylsilyl) trifluoroacetamide (BSTFA, with 1 % trimethylchlorosilane as catalyst) was added to react at 75 °C for 70 min. The derivatives were then diluted with n-hexane to 1 mL and immediately analyzed by a Shimadzu TQ8040 gas chromatography-mass spectrometry (GC-MS) in



electron ionization (EI) mode. A RXi-5SilMS capillary column (30 m × 0.25 mm i.d., film thickness 0.25 μm) was used as the GC column and helium was used as the carrier gas (1.0 mL min$^{-1}$). The injector was set splitless at an temperature of 290 °C. The programmed oven temperature increased from 70 °C to 150 °C at 2 °C min$^{-1}$, then to 200 °C at 5 °C min$^{-1}$, then to 300 °C at 25 °C min$^{-1}$, and stay at 300 °C for 6 min. The organic tracers were quantified using the calibration curves of the derivatives of authentic standards, which were obtained right before the measurement of the ambient samples.

The recovery rates of the measured organic tracers were determined by measuring the authentic standards spiked onto the pre-baked blank quartz filters following the same procedure as the ambient samples. The recovery rates of the measured organic tracers were in the range of 70-110 %, except for 4-methyl-5-nitrocatechol, which showed recoveries of 36-57 % with an average of 47.8 %. The relative standard deviation (RSD) of the measurement of 4-methyl-5-nitrocatechol was 18.0 % based on four repeated measurements, which met the analysis requirement of environmental samples (RSD<30 %).

Therefore, the concentration of 4-methyl-5-nitrocatechol was corrected for recovery, while that of other organic tracers was not. The blank filters found no significant contamination (<10 % of the concentration in ambient samples), and the final ambient concentrations were corrected for blank.

To measure the contents of water-soluble organic carbon (WSOC) and water-soluble inorganic ions, one-fourth of each sample filter was cut into pieces and ultrasonically extracted with 35 mL Milli-Q water (> 18.2 MΩ) for 40 min. The extract

was filtered through a 0.45 μm PTFE syringe filter and split into two portions. One portion was used to quantify WSOC by a total organic carbon (TOC) analyzer (Shimadzu TOC-L CPN), and the other was used to measure the four anions (Cl$^-$, NO$_3^-$, SO$_4^{2-}$, C$_2$O$_4^{2-}$) and five cations (Na$^+$, NH$_4^+$, K$^+$, Mg$^{2+}$, Ca$^{2+}$) by a Dionex 600 ion chromatography. In addition, the concentrations of organic carbon (OC) and elemental carbon (EC) were analyzed by a DRI 2001A carbon analyzer following the IMPROVE thermal/optical reflectance (TOR) protocol.

**2.3 Source apportionment by PMF**

EPA PMF 5.0 was used to quantify the contributions of primary and secondary sources to WSOC as well as OC in aerosols of different sizes. A total of 16 species were chosen as the inputs of PMF, including WSOC, OC, EC, SO$_4^{2-}$, NO$_3^-$, NH$_4^+$, C$_2$O$_4^{2-}$, Ca$^{2+}$, Mg$^{2+}$, levoglucosan, cholesterol, 4-methyl-5-nitrocatechol, phthalic acid, 2-methylerythritol, 3-hydroxyglutaric acid, and *cis*-pinonic acid. The concentration uncertainties of the target species were calculated as follow:

$Unc = 5/6 \times MDL \ (c \leq MDL)$        (1)

$Unc = \sqrt{(P \times c)^2 + (0.5 \times MDL)^2} \ (c > MDL)$        (2)

where MDL is the minimum detection limit of the measuring instrument, P is the measurement error fraction, and c is the concentration of target species. The measurement error fraction was set to 10 % by experience for WSOC, OC, EC, SO$_4^{2-}$, NO$_3^-$, NH$_4^+$, C$_2$O$_4^{2-}$, Ca$^{2+}$, and Mg$^{2+}$ (Gao et al., 2014; Yang et al., 2016b). The error fractions of organic tracers were

estimated by the relative standard deviation of repeated tests: the values for levoglucosan, cholesterol, 2-methylerythritol, 3-hydroxyglutaric acid and phthalic acid were set to 10 %, and the values of 4-methyl-5-nitrocatechol and *cis*-pinonic acid



were set to 15 %. The PMF model was run repeatedly with minor adjustment of factor numbers and uncertainties to get the optimal solution.

### 2.4 Aerosol water content calculation and other supplementary data collection

To explore the formation mechanism of secondary organic tracers, the inorganic aerosol water content (AWC) in $PM_1$, $PM_{2.5}$ and $PM_{10}$ was estimated by the reverse mode calculation of ISORROPIA-II model, which is a computationally efficient thermodynamic equilibrium model for inorganic aerosols (Fountoukis and Nenes, 2007). The contribution of organic compounds to AWC was estimated by the same approach employed by Cheng et al. (2016). The total AWC was the sum of the water content in inorganic and organic aerosols.

The hourly concentration data of $PM_{2.5}$, $PM_{10}$, $O_3$, and $NO_2$ that were measured at an urban air quality monitoring station (39.89° N, 116.38° E) were obtained online (http://www.envicloud.cn). The meteorological data including temperature (T), relative humidity (RH), wind speed (WS), wind direction (WD), solar radiation (SR) and precipitation were recorded at the sampling site using an automatic meteorological station. The average temperature, relative humidity, and wind speed was 2.1 °C, 64.2 %, and 0.85 m s$^{-1}$, respectively for the winter period, and 12.5 °C, 64.9 % and 0.89 m s$^{-1}$ for the spring periods.
One precipitation process occurred from the night of March 22nd to the night of March 24th during the whole sampling period, and the total rainfall was 13.4 mm.

## 3 Results and discussion

### 3.1 Distribution characteristics of WSOC in $PM_1$, $PM_{2.5}$ and $PM_{10}$ during haze episodes

The temporal variations of WSOC, OC and particulate mass concentrations in $PM_1$, $PM_{2.5}$ and $PM_{10}$ in Beijing during the
whole sampling period are shown in Fig. 1 and the average concentrations of the identified species in $PM_1$, $PM_{2.5}$ and $PM_{10}$ during haze episodes with $PM_{2.5}$ higher than 75 μg m$^{-3}$ are provided in Table 1. Compared to the haze episodes in spring, the haze episode in winter was characterized by higher particulate mass concentrations and more stagnant weather conditions as indicated by lower wind speed, lower temperature, and higher relative humidity. Compared to the total particulate matter, the mass concentrations of WSOC and OC in the corresponding particulate matter showed stronger seasonal variations, and the
average mass concentrations of WSOC and OC during the haze episode in winter were around three times of those in spring (Table 1). Both WSOC and OC were enriched in $PM_{2.5}$ during the haze episode in winter with comparable mass concentrations in the size ranges of 0-1 μm and 1-2.5 μm, while the proportions in both finer (0-1 μm) and coarse particles (2.5-10 μm) increased in spring.

As secondary organic aerosol takes a large proportion of WSOC, the WSOC/OC ratio can be used to infer the extent of
secondary formation and/or aging of aerosols (Ram et al., 2012). As shown in Fig. 1, the WSOC/OC ratio remained relatively stable during polluted days and the ratio sharply dropped when wind or rain cleansed the atmosphere of particulate




matter. Compared to the WSOC/OC ratios reported in the literature (Table S1), the average WSOC/OC ratio during the haze episodes in this study are significantly higher than the corresponding seasonal averages in Beijing, which was probably attributable to the enhanced SOA production during haze days (Cheng et al., 2013a; Zhou et al., 2014). The severe SOA

pollution during the haze episodes was also evidenced by the high OC/EC ratios as shown in Table 1. The WSOC/OC ratio in spring was obviously higher than that in winter, indicating higher proportion of secondary or more aged aerosols in spring. Similar seasonal variation of WSOC/OC was also reported in the literature (Table S1), which was probably due to the enhanced production of secondary aerosols at higher temperature (Jaffrezo et al., 2005; Xiang et al., 2017). As also shown in Table 1, the WSOC/OC ratio was higher during the day than that at night, which was consistent with the stronger

photochemical processes and enhanced secondary formation during the day. The WSOC/OC ratios in $PM_1$, $PM_{2.5}$ and $PM_{10}$ were very similar during the haze episode in winter, while the ratios in $PM_1$, $PM_{2.5}$ and $PM_{10}$ showed greater difference in spring with the general order of $PM_1 > PM_{2.5} > PM_{10}$.

## 3.2 Distribution and chemical characteristics of organic tracers

### 3.2.1 Mass concentrations of organic tracers during haze episodes

The average mass concentrations of the identified organic tracers in $PM_1$, $PM_{2.5}$ and $PM_{10}$ during haze episodes with $PM_{2.5}$ higher than 75 μg m$^{-3}$ are also shown in Table 1. The identified organic tracers covered three major source categories of WSOC, including primary sources, anthropogenic SOA and biogenic SOA. Levoglucosan and cholesterol are primary WSOC tracers for biomass burning and cooking, respectively. Phthalic acid and 4-methyl-5-nitrocatechol are anthropogenic SOA tracers for aromatic SOA and biomass burning SOA, respectively (Iinuma et al., 2010; Al-Naiema and Stone, 2017). 2-

methylerythritol, 3-hydroxyglutaric acid and *cis*-pinonic acid are biogenic SOA tracers with 2-methylerythritol acting as isoprene SOA tracer and 3-hydroxyglutaric acid and *cis*-pinonic acid as monoterpene SOA tracers. Compared to the reported values of organic tracers in the literature, the average mass concentrations of primary WSOC tracers and anthropogenic SOA tracers (levoglucosan, cholesterol and phthalic acid) in $PM_{2.5}$ during the haze episodes were 0.7-13.2 times higher than the corresponding seasonal averages in Beijing (He et al., 2006; Tao et al., 2016; Zhao et al., 2018), while the average mass

concentrations of biogenic SOA tracers (2-methylerythritol, 3-hydroxyglutaric acid, *cis*-pinonic acid) in $PM_{2.5}$ during the haze episodes in winter and spring were 0.4-6.2 times lower than those in summer in urban Beijing (Yang et al., 2016a).

As shown in Table 1, both the primary WSOC tracers and the anthropogenic SOA tracers in $PM_1$, $PM_{2.5}$ and $PM_{10}$ showed elevated mass concentrations during the haze episode in winter compared to those in spring. In addition, the average mass concentrations of the anthropogenic SOA tracers (phthalic acid and 4-methyl-5-nitrocatechol) in winter were around 5 times

of those in spring while the average mass concentrations of the primary WSOC tracers (levoglucosan and cholesterol) in winter were around 2.5 times of those in spring, showing enhanced SOA formation from anthropogenic precursors during the haze episode in winter. Both the low atmospheric mixing height during the haze episode in winter and the additional emissions of anthropogenic precursors (such as polycyclic aromatic hydrocarbons) associated with domestic heating resulted

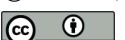



in the accumulation of anthropogenic precursors and consequently the anthropogenic SOA. On the contrary, the average mass concentrations of monoterpene SOA tracers (*cis*-pinonic acid and 3-hydroxyglutaric acid) in spring were higher than those in winter, which probably resulted from the enhanced biogenic monoterpene emissions and SOA formation at higher temperature (Cheng et al., 2018). However, different from the monoterpene SOA tracers, the isoprene SOA tracer, 2-methylerythritol, showed higher concentration in winter. Previous studies reported emissions of large amounts of isoprene from all biomass burning types (Akagi et al., 2011; Li et al., 2018). Therefore, the intense biomass burning activities in winter as evidenced by the high concentration of levoglucosan may contribute to the enhanced concentration of isoprene as well as its SOA tracer in winter. In addition, previous studies also showed that sulfate could increase the solubility of isoprene-derived epoxydiols (IEPOX) in the aqueous phase of aerosols through salting-in effect, and promote the ring-opening reaction of IEPOX and the subsequent isoprene SOA formation through nucleophilic attack (Xu et al., 2015; Li et al., 2018). Therefore, the higher concentration of 2-methylerythritol during the haze episode in winter may also be attributable to the enhanced SOA formation from isoprene in the presence of higher concentration of sulfate as shown in Table 1.

### 3.2.2 Diurnal patterns and size distributions

The diurnal patterns of the organic tracers in $PM_1$, $PM_{2.5}$ and $PM_{10}$ during the haze episodes ($PM_{2.5} > 75$ μg m$^{-3}$) in winter and spring are shown in Fig. 2. Compared to the nighttime, the daytime atmosphere is typically characterized by higher atmospheric mixing height and stronger source emissions & atmospheric photochemical activities, which exert opposite effects on the diurnal patterns of atmospheric pollutants. For example, the WSOC concentration was slightly higher at night than that by day in winter whereas the diurnal pattern of WSOC was opposite in spring (Table 1), which was probably due to the different dominating atmospheric processes in winter and spring. As shown in Fig. 2 and Table 1, the concentrations of the primary WSOC tracers and anthropogenic SOA tracers were much higher at night in winter, whereas the daytime and nighttime concentrations were similar in spring due to the smaller diurnal difference of atmospheric mixing height and stronger daytime photochemical activities in spring compared to winter. However, levoglucosan (primary biomass burning tracer) in aerosols of all sizes and 4-methyl-5-nitrocatechol (SOA tracer from biomass burning) in $PM_{2.5}$ and $PM_{10}$ were exceptions, which showed significantly higher concentrations at night in spring. It was speculated that the greatly enhanced nighttime concentrations of levoglucosan and 4-methyl-5-nitrocatechol in spring resulted from the uncontrolled burning activities of biomass at night. Similar phenomenon has also been reported in our previous study (Yang et al., 2016b). The monoterpene SOA tracer *cis*-pinonic acid showed higher concentrations in the daytime in both seasons with greater diurnal variation in spring compared to winter, indicating the dominant effect of photochemical oxidation and secondary formation by day. In contrast, another monoterpene SOA tracer 3-hydroxyglutaric acid showed similar daytime and nighttime concentrations in both seasons and the isoprene SOA tracer 2-methylerythritol showed higher concentration at night especially in spring. Compared to *cis*-pinonic acid, 2-methylerythritol and 3-hydroxyglutaric acid are higher-generation oxidation products of biogenic precursors and are formed in longer time scales (Fu et al., 2010). As such, the direct



promoting effect of photochemical activities during the daytime was weaker on them than that on *cis*-pinonic acid. Besides, the enhanced 2-methylerythritol at night in spring was probably due to the enhanced emission of isoprene from biomass burning, which was consistent with the diurnal patterns of levoglucosan and 4-methyl-5-nitrocatechol in spring. Except for

*cis*-pinonic acid and 3-hydroxyglutaric acid, the identified organic tracers showed the least diurnal difference in $PM_1$ compared to $PM_{2.5}$ and $PM_{10}$.

Most of the identified organic tracers showed similar size distributions with WSOC with enrichment in $PM_{2.5}$ and comparable mass concentrations in the size ranges of 0-1 μm and 1-2.5 μm during the haze episode in winter and increased proportions in finer (0-1 μm) or coarse particles (2.5-10 μm) in spring (Fig. 2 and Table 1). *Cis*-pinonic acid was an

exception and was mainly distributed in $PM_1$ during all the haze episodes in winter and spring, consistent with the previously reported results (Kavouras and Stephanou, 2002; Herckes et al., 2006). The enrichment of *cis*-pinonic acid in $PM_1$ indicates that *cis*-pinonic acid in atmospheric aerosols was formed through gas-phase photochemical reactions and nucleation, which results in the accumulation in finer particles (Yu et al., 1999). As for other SOA tracers including 4-methyl-5-nitrocatechol, phthalic acid, 2-methylerythritol, and 3-hydroxyglutaric acid, a considerable fraction of them were distributed in the size

range of 1-10 μm especially 1-2.5 μm, which possibly resulted from the hygroscopic growth of aerosols and the facilitated aqueous-phase oxidation on the surface of particles during haze episodes. Besides, the proportions of these SOA tracers in the size range of 1-10 μm were higher at night than by day, and higher in winter than in spring, implying higher contributions of aerosol hygroscopic growth and aqueous-phase oxidation to the formation and distribution of these SOA tracers at night and in winter.

**3.2.3 Influencing factors and possible formation mechanisms of SOA tracers**

To explore the influencing factors of SOA tracers, the correlation coefficients between SOA tracers and several meteorological parameters, $O_3$, aerosol acidity [$H^+$], and aerosol water content (AWC) in $PM_1$, $PM_{2.5}$ and $PM_{10}$ during the whole sampling period are listed in Table 2. The comparison of the correlation coefficients among different SOA tracers in $PM_{2.5}$ is shown in Fig. 3. Overall, the anthropogenic SOA tracers 4-methyl-5-nitrocatechol and phthalic acid exhibited

relatively strong positive correlations with aerosol water content, aerosol acidity and relative humidity and relatively strong negative correlations with wind speed, temperature, solar radiation and $O_3$. Typically, higher wind speed, higher temperature, and lower relative humidity are associated with higher atmospheric mixing height, thereby are favorable for the dispersion of atmospheric pollutants (Zhang et al., 2017). Besides, the strong negative correlations of the anthropogenic SOA tracers with $O_3$ and solar radiation as well as temperature suggest that gas-phase photo-oxidation was not the major formation mechanism

for 4-methyl-5-nitrocatechol and phthalic acid during the haze episodes. Instead, the secondary formation was enhanced by aqueous-phase oxidation and aerosol acidity during the haze episodes as evidenced by the strong positive correlations with aerosol water content, aerosol acidity and relative humidity. Chamber studies also showed that increasing particle water content could significantly enhance the aromatic SOA yield (Zhou et al., 2011).


Compared to the anthropogenic SOA tracers, the monoterpene SOA tracer *cis*-pinonic acid showed distinctively different
correlations with the factors under consideration. Overall, *cis*-pinonic acid showed strong positive correlation with
temperature, moderate positive correlation with solar radiation, $O_3$ and aerosol acidity, and poor correlation with aerosol
water content, relative humidity and wind speed. Previous studies also showed that higher temperature could enhance
monoterpene emission and the subsequent SOA formation (Ding et al., 2011; Shen et al., 2015). Consistent with the
formation mechanism inferred from the diurnal pattern and size distribution of *cis*-pinonic acid (Sec. 3.2.2), the correlation
pattern of *cis*-pinonic acid with different influencing factors further proved that the gas-phase photochemical oxidation was
the major formation pathway of *cis*-pinonic acid. As shown in Fig. 3, compared with *cis*-pinonic acid, aerosol water content
and relative humidity became more important factors influencing the concentrations of 3-hydroxyglutaric acid and 2-
methylerythritol while their correlations with temperature, solar radiation, and $O_3$ became weaker or even negative.
Therefore, it was inferred that the formation of 3-hydroxyglutaric acid and 2-methylerythritol was a combination of the gas-
phase and aqueous-phase oxidation processes with 3-hydroxyglutaric acid more on the gas-phase oxidation side and 2-
methylerythritol more on the aqueous-phase oxidation side.

The formation mechanism of 2-methylerythritol from isoprene has been extensively studied in the literature. Basically, the
production of 2-methylerythritol from isoprene can be achieved through two pathways, including the gas-phase oxidation
with OH radical and the acid-catalyzed multiphase reactions with hydrogen peroxide (Claeys et al., 2004a; Claeys et al.,
2004b). The formation of 2-methylerythritol can be enhanced by the reactive uptake and subsequent aqueous-phase
oxidation of the isoprene-derived epoxydiols (IEPOX) formed in the gas phase (Surratt et al., 2010; Xu et al., 2015; Riva et
al., 2016). Previous study showed that the concentration of 2-methyltetrols increased with temperature rapidly on days with
temperature higher than 20 °C (Liang et al., 2012). However, the temperature range covered in this study was relatively low
with the average of 2.1 °C in winter and 12.5 °C in spring. Therefore, the enhancement of the gas-phase oxidation at higher
temperature in spring was probably masked by other factors such as the decreased rate of aqueous-phase oxidation and
reduced emissions from biomass burning.

The correlation pattern of WSOC/OC with different influencing factors was similar with that of 3-hydroxyglutaric acid, and
relative humidity and aerosol water content appeared to be crucial factors affecting the WSOC/OC ratio (Table 2). The
strong positive correlations with RH and aerosol water content were attributable to two facts: (1) the gas-phase WSOC could
more easily partition into the aerosol phase at higher aerosol water content (Hennigan et al., 2009); (2) the aqueous-phase
production of WSOC could be enhanced at higher aerosol water content (Du et al., 2014; Xiang et al., 2017). The correlation
patterns of the SOA tracers with different influencing factors show slight difference among $PM_1$, $PM_{2.5}$ and $PM_{10}$. One
general rule that can be reached was that the correlation coefficients of 4-methyl-5-nitrocatechol, phthalic acid, 2-
methylerythritol, and 3-hydroxyglutaric acid with relative humidity and aerosol water content increased with particle size,
indicating higher contributions of the aqueous oxidation process to the formation of these compounds in larger aerosols
following the hygroscopic growth of aerosols during haze episodes.



### 3.3 Source apportionment of WSOC in PM$_1$, PM$_{2.5}$ and PM$_{10}$ during haze episodes

#### 3.3.1 Source apportionment by PMF

The primary and secondary sources of WSOC were quantified by the PMF 5.0 model with WSOC, EC, SO$_4^{2-}$, NO$_3^-$, NH$_4^+$,
C$_2$O$_4^{2-}$, Ca$^{2+}$, Mg$^{2+}$, and the seven organic tracers as the inputs of PMF. Theoretically, the source profiles of WSOC would be
different in different seasons and for particles with different sizes. However, considering the sample size requirement by
PMF (preferably more than 100 samples), we assumed that the source profile of WSOC was identical in different seasons
and for particles with different sizes and used the whole-period concentration data as one input into the PMF model to obtain
the source profile of WSOC. Such simplification was justified as the inputs of the source profile were carefully selected
representative source tracers.

As shown in Fig. 4, nine factors for the source apportionment of WSOC were identified in this study, including four primary
sources and five secondary sources. Factors 1 and 2 were characterized by a high level of levoglucosan and cholesterol
respectively, thus were identified as primary emissions from biomass burning and cooking respectively. Factor 3 showed a
high loading of EC that could not be interpreted by biomass burning, indicating emissions from other primary combustion
sources such as coal combustion. High levels of Ca$^{2+}$, Mg$^{2+}$ were observed in Factor 4, which was thus interpreted as dust
source. Factors 5 and 6 were characterized by a high level of 4-methyl-5-nitrocatechol and phthalic acid respectively, thus
were identified as secondary organic carbon (SOC) from biomass burning and aromatic compounds respectively. Factor 7
showed a high loading of 2-methylerythritol, and was identified as SOC from isoprene. Factor 8 showed high levels of *cis*-
pinonic acid and 3-hydroxyglutaric acid, and was identified as SOC from monoterpenes. Factor 9 exhibited a strong link
with SO$_4^{2-}$, NO$_3^-$, NH$_4^+$ and C$_2$O$_4^{2-}$, suggesting a secondary nature of this factor, thus was interpreted as SOC from other
sources.

Source contributions to WSOC in PM$_1$, PM$_{2.5}$ and PM$_{10}$ in Beijing during the sampling periods in winter and spring are
presented in Fig. 5. The contributions of secondary sources to WSOC were more than 50 % in both winter and spring. The
major sources of WSOC in winter were other primary combustion sources and aromatic SOC, and their contributions
decreased significantly in spring when the contributions of other SOC and biogenic SOC obviously increased. Domestic
heating was probably the major cause of the enhancement of other primary combustion sources and aromatic SOC in winter.
According to the results of previous studies, biomass burning contributed to more than 30 % of WSOC in PM$_{2.5}$ in winter
and was therefore the dominant source of wintertime WSOC in Beijing (Cheng et al., 2013b; Du et al., 2014; Tao et al.,
2016). In contrast, the contribution of biomass burning (Factor 1 + Factor 5) to WSOC in PM$_{2.5}$ in the current study was
22.1 % and 17.9 % in winter and spring respectively, much lower than the previously reported results. The decreased
contribution of biomass burning was possibly due to the effective control of biomass burning around Beijing in recent years,
or it might also indicate the enhanced contributions of other sources during haze episodes compared to the non-haze days.
Moreover, the contribution of biomass burning to WSOC was more in the form of secondary formation rather than primary



emission in winter while primary emission became the dominant form in spring, indicating different burning pattern and
secondary reaction mechanism in winter and spring. Cooking and dust contributed the least to WSOC, while their
contributions were both increased in spring compared to winter.

Comparing the source contributions of WSOC in $PM_1$, $PM_{2.5}$ and $PM_{10}$, it can be found that the contribution of the sum of
primary sources to WSOC followed the order of $PM_1 > PM_{2.5} > PM_{10}$ in winter, implying that primary sources have greater
influence on finer particles. Higher correlation between WSOC and POC (primary organic carbon) in $PM_1$ than that in $PM_{2.5}$
was also observed during haze episodes in Shanghai (Qiao et al., 2016). The increased contribution of secondary sources in
coarser particles during haze episode in winter was likely associated with the hygroscopic growth of aerosols and the
enhanced secondary formation through aqueous phase oxidation. The contributions of the total primary and secondary
sources to WSOC in spring were around 40 % and 60 % respectively. While the primary and secondary contributions were
similar in $PM_1$, $PM_{2.5}$ and $PM_{10}$ in spring, the contributions of the respective sources were different. In particular, the
contribution of other primary combustion sources to WSOC decreased in larger particles in spring while that of biomass
burning obviously increased.

### 3.3.2 Comparison of SOC estimated using different methods

Quantifying SOA in the atmosphere has been difficult and several methods have been used to roughly estimate the amounts
of SOA, including the OC-EC method, WSOC-levoglucosan method, and PMF-based methods. To evaluate the potential
error of different SOC estimation methods, SOC estimated by different methods were compared. The estimation of SOC by
the OC-EC method was calculated by $SOC_{OC-EC} = OC - (OC/EC)_{min} \times EC$, where $(OC/EC)_{min}$ is the minimum OC to EC ratio
during the sampling period, representing the OC to EC ratio from primary emissions (Lim and Turpin, 2002). The minimum
OC to EC ratios in $PM_1$, $PM_{2.5}$ and $PM_{10}$ during the whole sampling period were 1.84, 2.08, 2.03, respectively, which
appeared on the non-haze night of January 8th, 2017 with strong wind and low relative humidity. The WSOC-levoglucosan
method is based on the assumption that SOC is water-soluble and WSOC mainly derives from biomass burning and SOC.
Thus SOC was estimated by $SOC_{WSOC-Levo} = WSOC - (WSOC/Levo)_{BB} \times C_{Levo}$, where $(WSOC/Levo)_{BB}$ is the ratio of WSOC
to levoglucosan from biomass burning, and $C_{Levo}$ is the ambient concentration of levoglucosan (Ding et al., 2008b). The ratio
of 10 was used as the $(WSOC/Levo)_{BB}$ ratio in winter as suggested by Yan et al. (2015), whereas the ratio of 8 was used in
spring as suggested by Feng et al. (2013). In fact, the $(WSOC/Levo)_{BB}$ ratio would change with the types of biomass and the
burning conditions, thus could vary significantly in different locations and seasons. Besides, the $(WSOC/Levo)_{BB}$ ratio might
also be different in aerosols of different sizes. Hence, the rough estimation of $(WSOC/Levo)_{BB}$ would result in considerable
uncertainty in the estimation of SOC. Moreover, it is questionable that all SOC is soluble in water. In fact, large contribution
of water-insoluble secondary organic aerosols has been observed in urban environment, which was attributed to the reactions
of anthropogenic precursors (Favez et al., 2008; Sciare et al., 2011).





The PMF-based methods included the WSOC-PMF and OC-PMF methods with WSOC and OC as the total variable in the model respectively. The five secondary sources of WSOC/OC resolved by PMF were summed up to calculate SOC: $SOC_{WSOC/OC-PMF}$ = isoprene SOC + monoterpene SOC + aromatic SOC + biomass burning SOC + other SOC. Source apportionment of OC by PMF was performed in the same way as WSOC, and the results are shown in Figures S1 and S2. Comparing the source contributions of WSOC and OC, it was found that primary sources particularly other primary 350 combustion sources and dust contributed more to OC than to WSOC.

Fig. 6 and Table 3 show the comparison of secondary organic carbon in $PM_1$, $PM_{2.5}$ and $PM_{10}$ in Beijing estimated by different methods during the sampling periods in winter and spring, and Table S2 shows the correlation coefficients among the estimated SOCs. Generally, SOC estimated by different methods exhibited similar trends during the whole sampling period, and the correlations were particularly strong (R>0.89) in winter. In spring, however, the OC-EC method showed 355 poorer correlation with other methods, especially the PMF-based methods (0.40<R<0.79). The two PMF-based methods showed the highest correlation with each other (R>0.95). Overall, the estimated SOC from the OC-based methods (OC-EC and OC-PMF methods) were higher than that from the WSOC-based methods (WSOC-levoglucosan and WSOC-PMF methods) especially in winter, indicating substantial contribution of water-insoluble SOC to total SOC in the atmosphere. Therefore, the SOC value estimated from the WSOC-based methods at best described the secondary portion of water-soluble 360 carbon in the atmosphere; using such value to represent total SOC in the atmosphere would result in underestimation of SOC in the atmosphere. Comparatively, $SOC_{WSOC-Levo}$ was obviously higher than $SOC_{WSOC-PMF}$, probably due to the overlook of the contributions of other primary combustion sources by the WSOC-levoglucosan method and thus $SOC_{WSOC-Levo}$ was overestimated.

The OC-EC method is frequently used in the literature to estimate secondary organic aerosol concentration in the atmosphere 365 for its convenience. As shown in Fig. 6 and Table 3, $SOC_{OC-EC}$ was close to $SOC_{OC-PMF}$ in winter, while in spring, $SOC_{EC}$ was significantly lower than $SOC_{OC-PMF}$ in $PM_{2.5}$ and $PM_{10}$. Noting that $SOC_{OC-EC}$ did not meet the decreasing trend of $PM_{10} > PM_{2.5} > PM_1$ during some days and that the OC-EC method showed poorer correlations with other methods, we suspected that non-negligible errors existed in the measurement of OC/EC. It has been reported that the OC and EC values would be significantly different using different analytical methods, especially for the EC value (Cheng et al. 2011a). For example, 370 compared to the thermal-optical transmittance (TOT) method, the thermal-optical reflectance (TOR) method would obtain higher EC values and lower OC values, thus the estimated SOC values would be lower if the TOR method was used to measure OC and EC (Cheng et al. 2011; Xiang et al., 2017). Though the OC data was also used in the SOC estimation by PMF methods, $SOC_{OC-PMF}$ was estimated by based on SOA tracers rather than EC values, thus was less affected by the OC/EC measurement methods.



## 4 Conclusions

Based on the diurnal $PM_1$, $PM_{2.5}$ and $PM_{10}$ samples collected in Beijing during haze episodes in winter and early spring, WSOC, OC, EC, water-soluble ions and organic tracers were accurately measured to investigate the distribution characteristics and sources of WSOC. The values of WSOC, OC and WSOC/OC ratio were all much higher during the haze episodes in this study compared to the seasonal averages in Beijing, suggesting higher pollution level of organic compounds and enhanced secondary formation during the haze episodes. Due to the additional contribution from domestic heating and the more stagnant weather conditions in winter, the haze episode in winter showed elevated levels of WSOC and OC, around three times of those in spring. Both WSOC and OC were enriched in $PM_{2.5}$ during the haze episode in winter with comparable mass concentrations in the size ranges of 0-1 μm and 1-2.5 μm, while the proportions in both finer (0-1 μm) and coarse particles (2.5-10 μm) increased in spring.

Most of the identified organic tracers showed similar seasonal variation, diurnal change and size distributions with WSOC, with elevated concentrations and more significant diurnal difference in winter and enrichment in $PM_{2.5}$. The biogenic SOA tracer cis-pinonic acid was an obvious exception, which showed higher concentration in spring, higher concentration during the day, and major distribution in PM1. Based on the distribution characteristics and correlation patterns with the influencing factors, it was inferred that cis-pinonic acid was mainly formed through the gas-phase photochemical oxidation while other SOA tracers were closely associated with the aqueous-phase oxidation. The gas-phase photochemical oxidation was weakened during the haze episodes, whereas the aqueous-phase oxidation became the major pathway of SOA formation during the haze episodes, especially in winter, at night and for the coarser particles.

Secondary sources accounted for more than 50% of WSOC in both winter and spring. The major sources of WSOC in winter were other primary combustion sources and aromatic SOC, whereas the contributions of other SOC and biogenic SOC obviously increased in spring. Contrary to the existing research results, biomass burning was not the dominant source of WSOC in Beijing during haze episodes. Moreover, the contribution of biomass burning to WSOC was more in the form of secondary formation in winter while primary emission became the dominant form in spring, indicating different burning pattern and secondary reaction mechanism in winter and spring. Primary sources showed greater influence on finer particles during haze episode in winter; however, secondary sources became more important for coarser particles, possibly due to the enhanced secondary formation through aqueous phase oxidation following the hygroscopic growth of aerosols.

SOC estimated by the OC-EC method, WSOC-levoglucosan method, and PMF-based methods were comparable during the whole sampling period, and the following points need to be taken into account when estimating SOC. (1) Water-insoluble SOC should not be ignored. (2) The WSOC-levoglucosan method may overestimate the secondary portion of WSOC due to overlook of other primary combustion sources. (3) SOC estimated by the OC-EC method may be underestimated when the TOR method was used to measure OC and EC. (4) SOC estimated by the PMF model was less affected by the measuring



methods of OC and EC, and increasing the number of samples and SOA tracers would improve the accuracy of SOC estimation.

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



**Table 1.** Average concentrations of the identified species in $PM_1$, $PM_{2.5}$ and $PM_{10}$ during haze episodes ($PM_{2.5}$>75 µg m$^{-3}$) in winter and spring.

| Compounds | Winter | | | | | | | | | Spring | | | | | | | | |
|---|---|---|---|---|---|---|---|---|---|---|---|---|---|---|---|---|---|---|
| | $PM_1$ | | | $PM_{2.5}$ | | | $PM_{10}$ | | | $PM_1$ | | | $PM_{2.5}$ | | | $PM_{10}$ | | |
| | Day | Night | Total | Day | Night | Total | Day | Night | Total | Day | Night | Total | Day | Night | Total | Day | Night | Total |
| WSOC (µg m$^{-3}$) | 14.1 | 15.3 | 14.7 | 27.0 | 31.1 | 29.2 | 31.7 | 34.8 | 33.4 | 6.6 | 6.0 | 6.3 | 9.2 | 8.8 | 9.0 | 11.4 | 10.5 | 10.9 |
| OC (µg m$^{-3}$) | 22.1 | 24.4 | 23.4 | 40.4 | 47.4 | 44.1 | 48.3 | 55.4 | 52.1 | 9.1 | 8.8 | 9.0 | 12.7 | 12.8 | 12.8 | 16.6 | 16.6 | 16.6 |
| EC (µg m$^{-3}$) | 5.5 | 7.9 | 6.8 | 9.0 | 12.2 | 10.7 | 10.5 | 13.4 | 12.1 | 2.3 | 2.6 | 2.4 | 3.5 | 4.2 | 3.8 | 4.6 | 5.1 | 4.9 |
| WSOC/OC | 0.66 | 0.60 | 0.63 | 0.67 | 0.64 | 0.65 | 0.65 | 0.61 | 0.63 | 0.77 | 0.73 | 0.75 | 0.74 | 0.70 | 0.72 | 0.70 | 0.69 | 0.69 |
| OC/EC | 4.01 | 3.38 | 3.68 | 4.40 | 4.07 | 4.22 | 4.47 | 4.16 | 4.30 | 4.08 | 3.62 | 3.84 | 3.73 | 3.17 | 3.44 | 3.77 | 3.33 | 3.54 |
| Particulate matter (µg m$^{-3}$) | 157.1 | 168.6 | 163.2 | 277.6 | 306.6 | 293.1 | 348.6 | 379.4 | 365.0 | 111.6 | 105.9 | 108.6 | 152.6 | 152.3 | 152.4 | 215.2 | 217.7 | 216.5 |
| **Organic tracers (ng m$^{-3}$)** | | | | | | | | | | | | | | | | | | |
| Levoglucosan | 427.6 | 546.1 | 490.8 | 693.7 | 977.5 | 845.0 | 805.2 | 1123.2 | 974.8 | 151.1 | 210.1 | 182.9 | 265.2 | 443.0 | 361.0 | 361.2 | 560.8 | 468.6 |
| Cholesterol | 16.2 | 19.0 | 17.7 | 18.5 | 31.8 | 25.6 | 22.5 | 37.9 | 30.7 | 8.1 | 7.2 | 7.7 | 9.6 | 10.2 | 9.9 | 12.8 | 14.1 | 13.5 |
| 4-Methyl-5-nitrocatechol | 107.5 | 119.4 | 113.8 | 163.5 | 227.4 | 197.6 | 187.6 | 288.2 | 241.2 | 23.8 | 22.0 | 22.8 | 34.1 | 46.9 | 41.0 | 37.0 | 54.2 | 46.3 |
| Phthalic acid | 81.9 | 82.3 | 82.1 | 187.1 | 233.1 | 211.6 | 233.8 | 294.9 | 266.4 | 25.1 | 22.9 | 23.8 | 36.0 | 38.5 | 37.4 | 48.3 | 45.0 | 46.4 |
| 2-Methylerythritol | 3.9 | 3.9 | 3.9 | 7.1 | 8.0 | 7.6 | 10.5 | 11.8 | 11.2 | 2.5 | 2.6 | 2.6 | 3.4 | 4.3 | 3.9 | 4.3 | 6.3 | 5.3 |
| 3-Hydroxyglutaric acid | 3.7 | 3.1 | 3.4 | 8.6 | 8.8 | 8.7 | 10.4 | 10.8 | 10.6 | 5.5 | 5.1 | 5.3 | 10.2 | 11.6 | 11.0 | 12.7 | 12.8 | 12.8 |
| *Cis*-pinonic acid | 2.7 | 1.6 | 2.1 | 3.1 | 2.4 | 2.7 | 3.6 | 3.1 | 3.3 | 5.9 | 3.7 | 4.7 | 6.5 | 4.5 | 5.4 | 8.0 | 5.4 | 6.6 |
| **Water-soluble ions (µg m$^{-3}$)** | | | | | | | | | | | | | | | | | | |
| Nitrate, $NO_3^-$ | 22.85 | 17.75 | 20.13 | 40.49 | 35.89 | 38.04 | 47.72 | 43.74 | 45.60 | 17.90 | 16.28 | 17.06 | 31.72 | 29.42 | 30.52 | 37.57 | 35.24 | 36.36 |
| Sulfate, $SO_4^{2-}$ | 13.66 | 13.01 | 13.31 | 30.24 | 33.19 | 31.81 | 36.70 | 41.27 | 39.14 | 6.27 | 5.84 | 6.05 | 11.56 | 11.04 | 11.29 | 13.71 | 13.20 | 13.44 |
| Chloride, $Cl^-$ | 2.69 | 3.12 | 2.92 | 4.31 | 5.25 | 4.81 | 6.35 | 7.14 | 6.77 | 1.18 | 1.17 | 1.18 | 1.93 | 1.87 | 1.90 | 2.72 | 2.64 | 2.68 |
| Oxalate, $C_2O_4^{2-}$ | 0.65 | 0.58 | 0.61 | 1.00 | 0.99 | 0.99 | 1.20 | 1.28 | 1.24 | 0.37 | 0.34 | 0.35 | 0.60 | 0.61 | 0.60 | 0.73 | 0.76 | 0.75 |
| Ammonium, $NH_4^+$ | 12.01 | 10.26 | 11.08 | 21.44 | 22.22 | 21.85 | 24.85 | 26.01 | 25.47 | 7.10 | 6.71 | 6.90 | 11.91 | 11.44 | 11.67 | 13.15 | 12.43 | 12.78 |
| Potassium, $K^+$ | 1.06 | 1.06 | 1.06 | 1.84 | 2.05 | 1.95 | 2.17 | 2.40 | 2.29 | 0.67 | 0.67 | 0.67 | 1.17 | 1.13 | 1.15 | 1.37 | 1.31 | 1.34 |
| Calcium, $Ca^{2+}$ | 0.86 | 0.71 | 0.78 | 0.78 | 0.69 | 0.74 | 3.02 | 2.94 | 2.98 | 1.21 | 1.05 | 1.13 | 1.04 | 0.90 | 0.96 | 4.35 | 3.94 | 4.14 |
| Sodium, $Na^+$ | 0.65 | 0.57 | 0.61 | 0.75 | 0.75 | 0.75 | 1.92 | 1.52 | 1.71 | 0.32 | 0.27 | 0.30 | 0.39 | 0.37 | 0.38 | 0.78 | 0.74 | 0.76 |
| Magnesium, $Mg^{2+}$ | 0.16 | 0.12 | 0.14 | 0.20 | 0.18 | 0.19 | 0.59 | 0.56 | 0.57 | 0.15 | 0.12 | 0.13 | 0.17 | 0.14 | 0.15 | 0.47 | 0.44 | 0.46 |



**Table 2.** Spearman correlation coefficients between the SOA tracers and meteorological parameters, $O_3$, aerosol acidity ($H^+$), and aerosol water content (AWC) in $PM_1$, $PM_{2.5}$ and $PM_{10}$ during the whole sampling period [a].

| Compounds | Size | T | RH | WS | SR [b] | $O_3$ | $H^{+}$ [c] | AWC |
|---|---|---|---|---|---|---|---|---|
| 4-Methyl-5-nitrocatechol | $PM_1$ | -0.36[*] | 0.50[**] | -0.63[**] | -0.52[*] | -0.67[**] | 0.39[**] | 0.65[**] |
| | $PM_{2.5}$ | -0.34[*] | 0.52[**] | -0.67[**] | -0.48[*] | -0.66[**] | 0.36[*] | 0.68[**] |
| | $PM_{10}$ | -0.33[*] | 0.52[**] | -0.65[**] | -0.49[*] | -0.64[**] | 0.25 | 0.70[**] |
| Phthalic acid | $PM_1$ | -0.50[**] | 0.28 | -0.45[**] | -0.64[**] | -0.68[**] | 0.30 | 0.61[**] |
| | $PM_{2.5}$ | -0.52[**] | 0.39[**] | -0.57[**] | -0.64[**] | -0.69[**] | 0.58[**] | 0.70[**] |
| | $PM_{10}$ | -0.48[**] | 0.41[**] | -0.57[**] | -0.65[**] | -0.65[**] | 0.38[*] | 0.73[**] |
| 2-Methylerythritol | $PM_1$ | 0.04 | 0.47[**] | -0.40[**] | -0.20 | -0.18 | 0.03 | 0.55[**] |
| | $PM_{2.5}$ | -0.06 | 0.58[**] | -0.47[**] | -0.31 | -0.30[*] | 0.36[*] | 0.65[**] |
| | $PM_{10}$ | -0.11 | 0.63[**] | -0.52[**] | -0.45[*] | -0.37[*] | 0.37[*] | 0.68[**] |
| 3-Hydroxyglutaric acid | $PM_1$ | 0.59[**] | 0.14 | -0.16 | 0.27 | 0.37[*] | 0.25 | 0.28 |
| | $PM_{2.5}$ | 0.38[*] | 0.42[**] | -0.38[*] | 0.03 | 0.17 | 0.48[**] | 0.51[**] |
| | $PM_{10}$ | 0.32[*] | 0.48[**] | -0.39[*] | -0.03 | 0.12 | 0.26 | 0.53[**] |
| *Cis*-pinonic acid | $PM_1$ | 0.76[**] | -0.08 | -0.05 | 0.33 | 0.46[**] | 0.14 | 0.09 |
| | $PM_{2.5}$ | 0.72[**] | -0.10 | -0.02 | 0.42 | 0.48[**] | 0.37[*] | 0.10 |
| | $PM_{10}$ | 0.67[**] | -0.07 | -0.07 | 0.40 | 0.43[**] | -0.15 | 0.13 |
| WSOC/OC | $PM_1$ | 0.37[**] | 0.30[*] | 0.09 | -0.11 | 0.30[*] | -0.03 | 0.20 |
| | $PM_{2.5}$ | 0.27 | 0.51[**] | -0.25 | -0.07 | 0.10 | 0.39[**] | 0.50[**] |
| | $PM_{10}$ | 0.35[*] | 0.28[*] | 0.13 | 0.18 | 0.39[**] | 0.26 | 0.26 |

[a] A precipitation process occurred between the night of March 22nd and the night of March 24th, thus was excluded from the correlation analysis.

[b] Solar radiation at night was close to zero, thus only the daytime data were included in the correlation analysis.

[c] $[H^+] = 2[SO_4^{2-}] + [NO_3^-] + [Cl^-] + 2[C_2O_4^{2-}] - [Na^+] - [NH_4^+] - [K^+] - 2[Mg^{2+}] - 2[Ca^{2+}]$.

Level of significance: [*]: $p < 0.05$; [**]: $p < 0.01$.





**Figure 1**. Temporal variations of meteorological conditions and WSOC, OC and particulate mass concentrations in PM$_1$, PM$_{2.5}$ and PM$_{10}$ in Beijing during the whole sampling period.






**Figure 2**. Diurnal variations of the average mass concentrations of organic tracers in $PM_1$, $PM_{2.5}$ and $PM_{10}$ during the haze episodes ($PM_{2.5} > 75$ μg m$^{-3}$) in winter and spring.





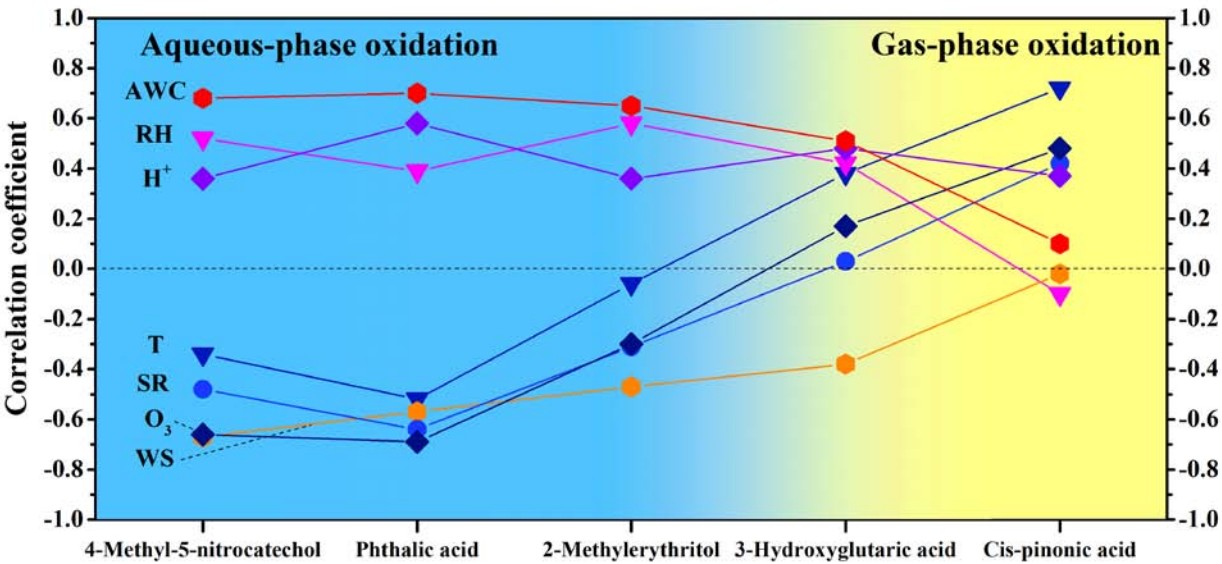

**Figure 3**. Comparison of the Spearman correlation coefficients with meteorological parameters, $O_3$, aerosol acidity ($H^+$), and
aerosol water content (AWC) in $PM_{2.5}$ among different SOA tracers.





**Figure 4**. Source profiles of WSOC in atmospheric particulate matter in Beijing resolved by PMF.





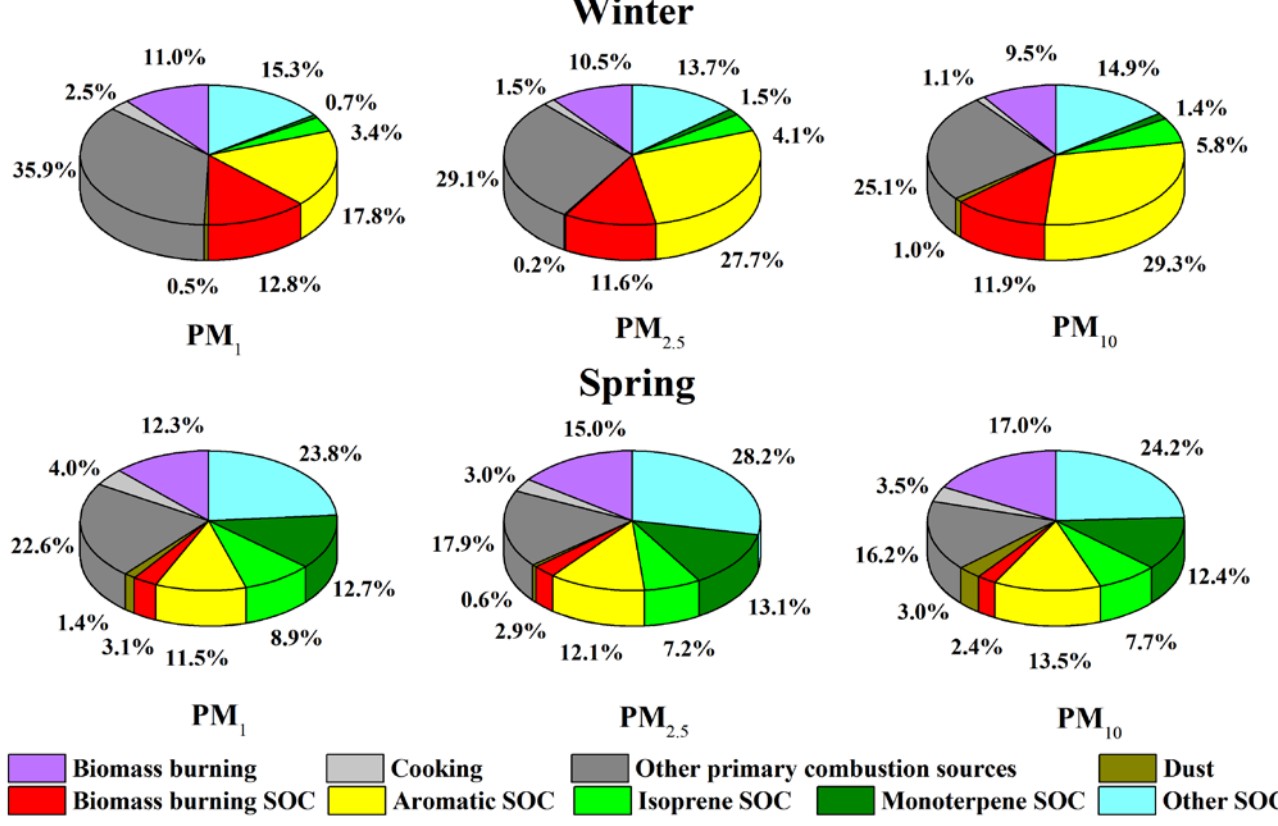

**Figure 5**. Source contributions to WSOC in PM₁, PM₂.₅ and PM₁₀ in Beijing during the sampling periods in winter and
spring.



**Figure 6**. Comparison of secondary organic carbon in PM$_1$, PM$_{2.5}$ and PM$_{10}$ in Beijing estimated by the OC-EC method, WSOC-levoglucosan method and PMF-based methods during the sampling periods in winter and spring.
