# Peer review of "Formation mechanism and source apportionment of water-soluble organic carbon in $PM_1$ , $PM_{2.5}$ and $PM_{10}$ in Beijing during haze episodes"

_Atmospheric Chemistry and Physics, 2018_

## Referee Comment (RC1) · Anonymous Referee #2 · 7 Oct 2018

This paper analyzed WSOC, EC, OC, inorganic ions and seven organic tracers in PM1, PM2.5 and PM10 filter samples collected in Beijing during the winter and spring of 2017. The contributions of primary versus secondary sources to WSOC were investigated by using PMF. The correlations of the organic tracers with meteorological parameters (e.g., temperature, relative humidity, solar radiation and wind speed) and ozone concentration were discussed. The objectives of this study include characterizing the distribution characteristics of WSOC in the different size of samples during the haze events in Beijing, quantitatively analyzing the contribution of different sources to WSOC, and elucidating the formation mechanism of WSOC in the city during the haze periods. WSOC is one of the major components of airborne particulate matter in the atmosphere, which poses significant impacts on climate change and human health. The topic of this manuscript is important, but the methodology and discussions are short of novelty and some discussions need to be clarified. In addition, this work only collected 10 days of samples (21 day/night samples in total ) in the winter and 15 days of samples (30 day/night samples in total ) in the spring to discuss size distribution, diurnal variation and seasonality of WSOC and related inorganic ions. The referee thinks such a small number of samples are not representative enough for discussing the aerosol chemistry seasonality. Following are the detailed comments, which should be addressed before this manuscript to be considered for publication by the journal.

1. Line 34-40, suspended soil dust is also an important source of WSOC.
2. Line 56-57, this is not true. In fact, many papers from China have reported WSOC and SOA formation, based on their field observation and lab simulation.
3. Line 70-75, how to define the haze episodes? Based on particle loading or visibility? Please clarify.
4. Line 70-82, as mentioned above, the number of samples collected in this study is small. Only ten days were selected for the winter and 15 days for the spring, why authors choose such a short time for the field measurement? How about the seasonal representativeness?
5. Line 83-85, why authors selected these seven organics as WSOC tracers to discuss WSOC sources and formation mechanisms? Here should give some explanation.
6. Line 95-102, the QA/QC, the recovery of 4-methyl-5-nitrocatechol is too low, which could cause a significant uncertainty for the measurement of 4-methyl-5-nitrocatecho concentration in real samples. Why only the concentration of 4-methyl-5-nitrocatechol was corrected by the recovery but others were not. If this difference in concentration correction could overestimate the importance of biomass burning and underestimate the contribution of other sources to WSOC?
7. Line 140-160, and also see Table 1. Why OC/EC ratio increased along with the increase of particle size in winter but decreased along with the particle size increase in spring?
8. Line 165-175, and also see Table 1, PM10-associated levoglucosan is on average two times that in PM1-associated, why? Levoglucosan is directly emitted from

biomass burning and usually stay in fine particles.

9. Line 198-210 and Figure 2 (i.e., Section 3.2.2). PM1-associated cholesterol showed a similar diurnal concentration in winter, but PM2.5- and PM10-associated cholesterol showed a concentration much higher in the nighttime than in the daytime, why?

10. Line 210-215. These statements are not reasonable. In fact, from Figure 2, we can see that *cis*-pinonic acid is higher in the daytime than in the nighttime, but other SOA tracers are higher at night, why?

11. Line 215-217 Authors should give the specific evidence to demonstrate that effect of photochemical activities during the daytime is weaker on 2-methylerythritol and 3-hydroxylglutaric acid than on *cis*-pinonic acid. Moreover, line 217-219, if the enhanced 2-methylerythritol concentration at night in spring was due to the enhanced biomass burning emissions, how about the correlation of 2-methylerythritol with levoglucosan?

12. Line 303-305 and also Figure 5, one may see that in winter SOC contributed to about 50%, 60% and 65% of PM1-associated, PM2.5-associated and PM10-assocciated WSOC, respectively. In other words, SOC was more enriched in 2.5-10 um size of coarse particles. Why?

---

## Referee Comment (RC2) · Anonymous Referee #1 · 21 Oct 2018

This paper investigated the size distribution and sources of WSOC in Beijing based on the offline PM1, PM2.5 and PM10 samples collected during haze episodes in winter and early spring of 2017. Many traditional chemical analysis are conducted in this study, which looks a lot of work. However, this study disadvantaged in the few sample numbers, the lack of in-depth data analysis or convincing data interpretation. This study may meet the minimum quality standard of ACP paper if all the following concerns are well addressed.

Major concerns:

1. With only ∼10 days measurement in each season, it was too board to attribute them

to "seasonal variations". The authors may want to limit the topic to two haze episodes in winter and spring. Or alternatively, the authors should state why they think these episodes are representative for winter/spring.

2. The PMF analysis require a lot of samples, usually at least 90. With the sample number shown in this study, any PMF result cannot be within the uncertainty range. The authors should avoid using good methods inappropriately.

3. It's surprising that the authors discussed about absolute tracer concentrations all the time. In winter and spring, the different boundary layer height and thus dilution effect could already result in the different pollutant accumulation rates, and thus much concentration differences. That is, even with the same emission rates, the primary tracers should also be higher in winter than in spring. The authors should consider scale primary and secondary tracers respectively to give a reasonable comparison.

4. In estimating the SOC through OC-EC methods, the primary OC/EC ratio should be estimated separately for each season, consider the seasonal variation in one major source of heating.

Minor suggestions:

1. Abstract and throughout: the term "diurnal samples" is strange. Consider change into "day/night sample pairs" or other clearer words.

2. Abstract and throughout: define "finer" and "coarser" modes clearly as PM1, PM1-2.5 or PM2.5-10.

3. Line 170: The difference between the two monoterpene SOA tracers, 3-hydroxyglutaric acid and cis-pinonic acid, should be clarified here.

4. Line 150 ∼ 152: The drop in WSOC/OC may not be due to the so-called "wind / rain cleansing", but is simply due to the change is air mass origins. The authors should investigate into the air mass origins during these days, and separate the influence of transport and chemistry.

5. Line 152 to 155: due to similar reasons above, higher WSOC/OC ratio during the haze episodes cannot represent the enhanced secondary formation, especially when the detailed POC/SOC contribution of WSOC is not known here. Does the SOC/OC ratio increase in haze episodes? A SOC/OC ratio versus PM1 / PM2.5 / PM10 concentrations can help better illustrate these issues.

6. Fig. 1 & Fig. 6: There should be gaps in lines when the dates are not continuous. In Fig. 6, plotting the corresponding PM concentrations together can also help make this figure clearer.

7. Fig. 2: use boxplots instead of averages is more proper here.

8. Fig. 3: Does the authors mean that only cis-pinonic acid is related to gas-phase oxidation, while all other tracers are attributed to heterogeneous reactions? Are there any references to support these findings? Also, does the tracers arranged according to the contribution of heterogeneous / gas-phase oxidation? If so, is the 3-hydroxyglutaric acid contributed by heterogeneous and gas-phase oxidation half and half? Please be more accurate in presenting figures.

9. Fig. 4 to 5 and relevant discussions: please double check whether one can get satisfactory PMF result from so few samples.

---

## Author Comment (AC1) · 10 Dec 2018

Reviewer #1:

Major concerns:

"1. With only ∼10 days measurement in each season, it was too board to attribute them to "seasonal variations". The authors may want to limit the topic to two haze episodes in winter and spring. Or alternatively, the authors should state why they think these episodes are representative for winter/ spring."

Response: We agree with the reviewer that instead of seasonal variation, our focus

was actually the haze episodes in winter and spring. The haze episodes investigated in this study occurred in January, late March and April, which were representative of haze episodes in winter and spring. Therefore, we have changed the term "seasonal variation" to "seasonal variation in the haze episodes in winter and spring" in the abstract and throughout the manuscript (Lines 16, 159, 467).

"2. The PMF analysis require a lot of samples, usually at least 90. With the sample number shown in this study, any PMF result cannot be within the uncertainty range. The authors should avoid using good methods inappropriately."

Response: We deeply appreciate the reviewer's concern on the sample number requirement of PMF analysis. It was true that the sources of PM1, PM2.5 and PM10 during the sampling periods in each season cannot be apportioned through individual PMF calculation due to limited sample number in each group. As a matter of fact, we used the whole-period concentration data (153 samples including PM1, PM2.5 and PM10 in both seasons) as one input into the PMF model to obtain a single source profile of WSOC. And the source contributions to particles with different sizes in each season were calculated and averaged in each group based on the single source profile of WSOC. In other words, a rough simplification was made that the source profile of WSOC was identical in different seasons and for particles with different sizes. Such simplification was justified as the inputs of the source profile were carefully selected representative source tracers. Besides, Han et al. (2006) showed that the source profiles were very similar in the resolved size ranges for the same source. Similar simplification has also been made in several other studies (Huang et al., 2006; van Drooge and Grimalt, 2015; Li et al. 2016; Tan et al., 2016; Tian et al., 2016; Wang et al., 2017). The appropriateness of the source profile result by PMF and the associated statistical parameters were taken into account to determine the optimal solution. As shown in Fig. 4 in the manuscript, the source profile that can clearly separate different source tracers was selected as the optimal solution. The Qtrue and Qrobust values of this solution agreed well, the distribution of residuals was close to normal with the calculated sum

of all species between -3 and +3, and the modeled concentration values correlated strongly (R2>0.94) with the measured values, indicating reliability of the selected PMF solution. More details of the PMF calculation process have been added in Sections 2.3 and 3.3.1 (Lines 131, 330) and the related references have also been added in the reference list.

"3. It's surprising that the authors discussed about absolute tracer concentrations all the time. In winter and spring, the different boundary layer height and thus dilution effect could already result in the different pollutant accumulation rates, and thus much concentration differences. That is, even with the same emission rates, the primary tracers should also be higher in winter than in spring. The authors should consider scale primary and secondary tracers respectively to give a reasonable comparison."

Response: We deeply appreciate the reviewer's valuable suggestion and have provided the relative concentrations of the organic tracers scaled by CO to better elucidate the difference in emission strength or secondary production rate in the haze episodes in winter and spring. The CO-scaled concentrations of the organic tracers are shown in Table 2 in the manuscript and the related discussion has been added in Section 3.2.1 (Line 206) as follows: The mass concentrations of the identified tracers in different seasons were not only controlled by the source emission strength or secondary formation rate, but also influenced by the different boundary layer height and thus dilution effect in different seasons. In comparison, carbon monoxide (CO) is mainly emitted from traffic exhaust with relatively stable emission strength in different seasons, and CO is generally inert to chemical reactions. Therefore, the variation of CO in the atmosphere can well reflect the influence of atmospheric physical processes such as dilution and mixing. To rule out the dilution effect and better elucidate the difference in emission strength or secondary production rate in the haze episodes in winter and spring, the relative concentrations of the identified organic tracers scaled by CO during haze episodes in winter and spring are shown in Table 2. The CO-scaled concentrations of levoglucosan and cholestrol in spring were significantly higher than in winter, indicating

stronger primary emissions of biomass burning and cooking in spring. The enchanced cooking emissions in spring might result from the uncontrolled outdoor barbecues as the weather became warmer (Hu et al., 2017). However, the CO-scaled concentrations of anthropogenic SOA tracers were significantly higher in winter, except for phthalic acid in PM1. Both the low atmospheric mixing layer height during the haze episode in winter and the additional emissions of anthropogenic precursors (such as polycyclic aromatic hydrocarbons) associated with domestic heating led to the accumulation of anthropogenic precursors and consequently the enhanced production rates of anthropogenic SOA tracers. The CO-scaled concentrations of biogenic SOA tracers were all much higher in spring than in winter, due to the enhanced biogenic emissions and subsequent SOA formation at higher temperature in spring (Shen et al., 2015). It was noteworthy that the enhancement of isoprene tracer (2-methylerythritol) was less significant than monoterpene tracers (3-hydroxyglutaric acid and cis-pinonic acid), though the observed isoprene concentration increased more significantly in Beijing in spring compared to monoterpenes (Cheng et al., 2018). Previous studies showed that sulfate could increase the solubility of isoprene-derived epoxydiols (IEPOX) in the aqueous phase of aerosols through salting-in effect, and promote the ring-opening reaction of IEPOX and the subsequent isoprene SOA formation through nucleophilic attack (Xu et al., 2015; Li et al., 2018). Thus the higher concentration of sulfate in winter as shown in Table 1 may facilitate the formation of isoprene SOA tracer in winter.

"4. In estimating the SOC through OC-EC methods, the primary OC/EC ratio should be estimated separately for each season, consider the seasonal variation in one major source of heating."

Response: We deeply appreciate the reviewer's valuable suggestion. As a matter of fact, we tried to use the minimum OC/EC ratio in each season to represent the primary OC/EC ratio. However, while the minimum OC to EC ratios in PM1, PM2.5 and PM10 (1.84, 2.08, 2.03 respectively) in winter were detected simultaneously on the non-haze night of January 8th, 2017 with strong wind and low relative humidity, the minimum OC

to EC ratios in PM1, PM2.5 and PM10 (2.08, 2.46, 2.61 respectively) in spring were detected at different time, on the daytime of March 24th, the night of March 15th and the night of March 22nd, respectively. Considering the relatively high concentrations of particulate matter during the time when minimum OC to EC ratios in PM1, PM2.5 and PM10 in spring were detected, secondary species might still be present in considerable amounts in the corresponding particulate matter. To avoid underestimation of SOC in spring, the minimum OC to EC ratios in PM1, PM2.5 and PM10 in winter were used to estimate SOC in both winter and spring. Typically, an OC/EC ratio of 2 is frequently used to represent the primary OC/EC ratio (Gray et al., 1986; Hildemann et al., 1991; Chow et al., 1996), which agreed well with the minimum OC/EC ratios detected in PM1, PM2.5 and PM10 in winter. We have added the above explanation on selection of primary OC/EC ratio in Section 3.3.2 (Line 380).

Minor suggestions:

"1. Abstract and throughout: the term "diurnal samples" is strange. Consider change into "day/night sample pairs" or other clearer words."

Response: We agree with the reviewer and the term "diurnal samples" has been changed to "day/night samples" throughout the revised manuscript.

"2. Abstract and throughout: define "finer" and "coarser" modes clearly as PM1, PM1-2.5 or PM2.5-10."

Response: The terms "finer" and "coarser" have been replaced by "PM1", "PM1-2.5", "PM1-10", or "PM2.5-10" in the revised manuscript where appropriate.

"3. Line 170: The difference between the two monoterpene SOA tracers, 3-hydroxyglutaric acid and cis-pinonic acid, should be clarified here."

Response: 3-hydroxyglutaric acid and cis-pinonic acid are the higher- and lower-generation oxidation products of monoterpenes respectively (Kourtchev et al., 2009). And the above statement has been added in the manuscript (Section 3.2.1, Line 195).

[Figure]

"4. Line 150~152: The drop in WSOC/OC may not be due to the so-called "wind /rain cleansing", but is simply due to the change in air mass origins. The authors should investigate into the air mass origins during these days, and separate the influence of transport and chemistry."

Response: We deeply appreciate the reviewer's valuable suggestion and have provided the backward trajectory analysis during the study periods in Section 3.1 (Line 165) as follows: Fig. 2 shows the cluster analysis results of 48-h backward trajectories during the sampling periods in winter and spring as well as the corresponding mean WSOC/OC ratio for each cluster. In winter, the air mass was along trajectory 1 in Dec. 31st 20:00 – Jan. 4th 8:00 and changed to trajectories 2 and 3 in Jan. 4th 8:00 – Jan. 8th 14:00, showing high and relatively stable WSOC/OC ratios in both periods, and the air mass was along trajectories 4 and 5 in Jan. 8th 14:00 – Jan. 11th 8:00, showing significantly lower WSOC/OC ratios. In spring, the air mass trajectories were more variable and mainly originated from the south (trajectories 1 and 5) and the northwest (trajectories 2 and 3), showing high WSOC/OC ratios. Trajectory 4 that occurred on Mar. 25th showed the lowest WSOC/OC ratio in spring. According to the backward trajectory analysis, the sharp drop of the WSOC/OC ratio at the end of the haze events were closely related to fresh and clean air masses, while the high and relatively stable WSOC/OC ratio during polluted days indicated contributions of secondary and/or aged aerosols.

"5. Line 152 to 155: due to similar reasons above, higher WSOC/OC ratio during the haze episodes cannot represent the enhanced secondary formation, especially when the detailed POC/SOC contribution of WSOC is not known here. Does the SOC/OC ratio increase in haze episodes? A SOC/OC ratio versus PM1 / PM2.5 / PM10 concentrations can help better illustrate these issues."

Response: We deeply appreciate the reviewer's valuable suggestion and have added the discussion on secondary formation based on the SOC/OC ratio in Section 3.3.3 (Line 430) as follows: Fig. 8 shows the temporal variations of relative contributions of

primary organic carbon (POC) and secondary organic carbon (SOC) in PM1, PM2.5 and PM10 during the sampling periods in winter and spring, estimated by the OC-PMF method. In general, POC dominated at the early stage of haze episodes and the contribution of SOC gradually increased with the evolution of haze episodes. Afterwards, POC dominated again when haze dissipated. The variation trend of the SOC/OC ratio tracked closely to that of relative humidity, especially for PM2.5 and PM10. To further elucidate the major formation mechanism of SOC during haze episodes, correlation analysis was conducted between the SOC/OC ratio and meteorological parameters, O3 concentration, particle mass concentration (M), aerosol acidity (H+) and aerosol water content (AWC) in PM1, PM2.5 and PM10 during the sampling periods in winter and spring respectively, as shown in Fig. 9. In winter, the SOC/OC ratio exhibited significantly positive correlations with relative humidity, aerosol water content and aerosol acidity and significantly negative correlations with solar radiation and O3 concentration, indicating that acid-catalyzed aqueous-phase reaction, rather than gas-phase photochemical oxidation, was the major formation mechanism of SOC during the haze episode in winter (Tang et al., 2018). The "dimming effect" caused by high particle concentrations during haze episodes led to the reduction of solar radiation and O3 concentration, which might explain the negative correlation of SOC/OC with SR and O3 (Zheng et al., 2015). The correlation coefficients increased with particle size in winter, suggesting that acid-catalyzed aqueous-phase reaction played a more important role on larger particles, which might result from the hygroscopic growth of aerosols and subsequent heterogeneous reactions on the surface of particles during haze episodes in winter (Tian et al., 2014; Ma et al., 2017; Xie et al., 2017). In addition, the SOC/OC ratio showed significantly negative correlations with wind speed, similar to the result reported in other studies (Ji et al., 2014; Shao et al., 2018). Stable meteorological conditions would result in accumulation of particles as well as the precursor gases, which could provide higher concentrations of reactants and abundant reaction media for heterogeneous reactions and lead to high SOC/OC ratio. In spring, the SOC/OC ratio showed weaker correlations with the influencing factors compared to winter. Besides, the correlation between SOC/OC and O3 turned positive in PM1 and PM10 in spring. The stronger solar radiation in spring might lead to higher contribution of gas-phase photochemical oxidation, thus the relative contribution of aqueous-phase oxidation was weakened. In summary, the acid-catalyzed aqueous-phase reaction could be the dominant formation mechanism of SOC during the haze episode in winter while the contribution of gas-phase photochemical oxidation increased in spring.

"6. Fig. 1 & Fig. 6: There should be gaps in lines when the dates are not continuous. In Fig. 6, plotting the corresponding PM concentrations together can also help make this figure clearer."

Response: Gaps in lines where the dates are not continuous have been added in Fig. 1 and Fig. 6 (Fig. 7 in the revised manuscript) and the corresponding PM concentrations have been added in Fig. 6 (Fig. 7 in the revised manuscript).

"7. Fig. 2: use boxplots instead of averages is more proper here."

Response: Plots in Fig. 2 (Fig. 3 in the revised manuscript) have been replaced by boxplots.

"8. Fig. 3: Does the authors mean that only cis-pinonic acid is related to gas-phase oxidation, while all other tracers are attributed to heterogeneous reactions? Are there any references to support these findings? Also, does the tracers arranged according to the contribution of heterogeneous / gas-phase oxidation? If so, is the 3-hydroxyglutaric acid contributed by heterogeneous and gas-phase oxidation half and half? Please be more accurate in presenting figures."

Response: Yes, the tracers in Fig. 3 (Fig. 4 in the revised manuscript) were arranged according to the contributions of heterogeneous / gas-phase oxidations. However, the contributions of heterogeneous and gas-phase oxidations could not be quantified in this study, and the tracer sequence in Fig. 3 (Fig. 4 in the revised manuscript) only roughly depicted the degree of correlation with aqueous-phase reaction and gas-phase oxida-

tion. Literature results have been added to support our findings as follow and addition references have be added in the reference list: Section 3.2.3, Line 279:... and phenolic compounds, which are emitted in significant amounts from biomass burning, could undergo fast reactions in aqueous-phase and produce substantial amounts of secondary organic aerosols (Sun et al., 2010; Li et al., 2014; Yu et al., 2014). Section 3.2.3, Line 295: As mentioned above, cis-pinonic acid and 3-hydroxyglutaric acid are the lower- and higher-generation products of monoterpenes respectively (Kourtchev et al. 2009). Chamber experiments have shown that cis-pinonic acid could be formed through gas-phase oxidation of monoterpenes (Yu et al., 1999), and field observations also suggested that cis-pinonic acid was closely linked to nucleation process as a first step in the formation of aerosols from organic vapors (O'Dowd et al., 2002; Laaksonen et al., 2008; Alier et al. 2013; van Drooge et al., 2018). However, 3-hydroxyglutaric acid, as the higher-generation product of monoterpenes, could be further formed through the aqueous-phase oxidation from the lower generation products. Aerosol mass spectrometer (AMS) observations also suggested that aqueous-phase processing exerts dominant impact on the formation of more oxidized SOA, while photochemical processing plays a major role in the formation of less oxidized SOA (Hu et al., 2017; Xu et al., 2017).

"9. Fig. 4 to 5 and relevant discussions: please double check whether one can get satisfactory PMF result from so few samples."

Response: Please refer to our response to Major Concern 2 for the justification of PMF calculation in this study.

Reviewer #2:

"1. Line 34-40, suspended soil dust is also an important source of WSOC."

Response: Thanks for the reviewer's suggestion and we have changed the statement on WSOC sources to "previous studies have also shown that primary emission sources other than biomass burning, such as vehicular exhaust emission, residual oil combustion, suspended soil dust, etc., also contribute to the WSOC load in the atmosphere (Kawamura and Kaplan, 1987; Huang et al., 2005; Guo et al., 2015; Kuang et al., 2015)" (Section 1, Line 38).

"2. Line 56-57, this is not true. In fact, many papers from China have reported WSOC and SOA formation, based on their field observation and lab simulation."

Response: Thanks for the reviewer's suggestion and we have changed the statement to "However, most of the field observations of WSOC focused on the seasonal variations on a whole-year basis, while the observations of SOA tracers were mainly conducted in summer with strong biogenic emissions; WSOC studies that focused on haze events were relatively few." (Section 1, Line 57).

"3. Line 70-75, how to define the haze episodes? Based on particle loading or visibility? Please clarify."

Response: Haze was defined by particle loading with 12-h PM2.5 above 75 $\mu$g m-3. And the above definition of haze has been added in Section 2.1 (Line 77).

"4. Line 70-82, as mentioned above, the number of samples collected in this study is small. Only ten days were selected for the winter and 15 days for the spring, why authors choose such a short time for the field measurement? How about the seasonal representativeness?"

Response: The main purpose of this study was to investigate the characteristics of WSOC in Beijing during haze episodes, which occurred most frequently in winter and spring. Instead of seasonal variation, our focus was actually the haze episodes in winter and spring. Therefore, the field measurements were limited to three haze episodes in winter and spring.

"5. Line 83-85, why authors selected these seven organics as WSOC tracers to discuss WSOC sources and formation mechanisms? Here should give some explanation."

Response: Thanks for the reviewer's suggestion and we have added explanation

of the seven selected organic tracers in Section 2.2 (Line 87) as follows: Levoglu-
cosan and cholesterol are primary WSOC tracers for biomass burning and cooking,
respectively. Phthalic acid and 4-methyl-5-nitrocatechol are anthropogenic SOA trac-
ers for aromatic SOA and biomass burning SOA, respectively (Iinuma et al., 2010;
Al-Naiema and Stone, 2017). 2-methylerythritol, 3-hydroxyglutaric acid and cis-pinonic
acid are biogenic SOA tracers with 2-methylerythritol acting as isoprene SOA tracer
and 3-hydroxyglutaric acid and cis-pinonic acid as monoterpene SOA tracers. 3-
hydroxyglutaric acid and cis-pinonic acid are the higher- and lower-generation oxidation
products of monoterpenes respectively (Kourtchev et al., 2009).

"6. Line 95-102, the QA/QC, the recovery of 4-methyl-5-nitrocatechol is too low, which
could cause a significant uncertainty for the measurement of 4-methyl-5-nitrocatechol
concentration in real samples. Why only the concentration of 4-methyl-5-nitrocatechol
was corrected by the recovery but others were not. If this difference in concentration
correction could overestimate the importance of biomass burning and underestimate
the contribution of other sources to WSOC?"

Response: Although the recovery of 4-methyl-5-nitrocatechol was low compared to
other organic tracers, the relative standard deviation (18.0 %) of the measurement
of 4-methyl-5-nitrocatechol was acceptable, confirming the reliability of the measure-
ment data. Therefore, the concentration of 4-methyl-5-nitrocatechol was corrected to
approach the true ambient concentration. Source contributions calculated by PMF
depend on the variation trends rather than the absolute concentrations of tracers;
the tracers that showed similar temporal variations with WSOC would be assigned
a higher contribution by PMF. For example, the absolute concentration of 4-methyl-5-
nitrocatechol was much higher than those of cis-pinonic acid and 3-hydroxyglutaric acid
in spring, however, the source contribution of biomass burning SOC by PMF was much
lower than that of monoterpene SOC. Concentration correction for recovery would not
change the temporal variation trend of 4-methyl-5-nitrocatechol, therefore the source
contribution of biomass burning SOC should not be overestimated by PMF calculations.

[Figure]

"7. Line 140-160, and also see Table 1. Why OC/EC ratio increased along with the increase of particle size in winter but decreased along with the particle size increase in spring?"

Response: As shown in Section 3.3.3 "Implications for secondary formation based on SOC/OC ratio", the percentage of secondary organic carbon in organic carbon increased with particle size in winter due to enhanced heterogeneous reactions, which partially explained why OC/EC ratio increased with particle size in winter. In spring, smaller difference of the percentage of secondary organic carbon in organic carbon was observed in particles of different sizes compared to winter, therefore, the decreasing OC/EC ratio with particle size can not be attributed to the variation of secondary organic carbon. Instead, the complex size distributions of OC and EC may result in the decreasing OC/EC ratio with particle size in spring. For example, EC showed a higher percentage in PM1-10 in spring than in winter, which would result in reduced OC/EC ratio in PM1-10 in spring.

"8. Line 165-175, and also see Table 1, PM10-associated levoglucosan is on average two times that in PM1-associated, why? Levoglucosan is directly emitted from biomass burning and usually stay in fine particles."

Response: Previous long-term observation in Beijing has reported the average annual ratio of 0.77 for the distribution of levoglucosan in PM2.5 and PM10 (Zhang et al., 2008). Although the PM10-associated levoglucosan was about two times that of PM1-associated, the average ratio of levoglucosan in PM2.5 and PM10 was 0.87 and 0.77 in winter and spring respectively, indicating the dominant distribution of levoglucosan in fine particles. The above statement has been added in Section 3.2.1 (Line 202).

"9. Line 198-210 and Figure 2 (i.e., Section 3.2.2). PM1-associated cholesterol showed a similar diurnal concentration in winter, but PM2.5-and PM10-associated cholesterol showed a concentration much higher in the nighttime than in the daytime, why?"

Response: We deeply appreciate the reviewer's attention to this detail. We found

it difficult to explain the results of cholesterol, thus we looked into the original raw data and found that the concentration of cholesterol in PM1 in winter was mistakenly calculated. The average concentration of cholesterol in PM1 in winter was 9.1 $\mu$g m-3 in the daytime and 19.0 $\mu$g m-3 in the nighttime, showing similar diurnal variation with that of PM2.5- and PM10-associated cholesterol. The average concentration of cholesterol in PM1 in winter has been corrected in Table 1.

"10. Line 210-215. These statements are not reasonable. In fact, from Figure 2, we can see that cis-pinonic acid is higher in the daytime than in the nighttime, but other SOA tracers are higher at night, why?"

Response: Yes, cis-pinonic acid was higher in the daytime than in the nighttime, but other SOA tracers were higher at night. The higher concentration of cis-pinonic acid in the daytime probably resulted from the enhanced and dominant contribution of photo-chemical oxidation and secondary formation by day (Yu et al., 1999). We have added the above statement in Section 3.2.2 (Line 244).

"11. Line 215-217 Authors should give the specific evidence to demonstrate that effect of photochemical activities during the daytime is weaker on 2-methylerythritol and 3-hydroxylglutaric acid than on cis-pinonic acid. Moreover, line 217-219, if the enhanced 2-methylerythritol concentration at night in spring was due to the enhanced biomass burning emissions, how about the correlation of 2-methylerythritol with levoglucosan?"

Response: The specific evidence to demonstrate that effect of photochemical activi-ties during the daytime is weaker on 2-methylerythritol and 3-hydroxylglutaric acid than on cis-pinonic acid has been provided in Section 3.2.3 (Line 295-312). It was inap-propriate to attribute the enhanced 2-methylerythritol concentration at night in spring to enhanced biomass burning emission because no significant correlation was found between 2-methylerythritol and levoglucosan. Therefore, we have deleted the corre-sponding statement "Besides, the enhanced 2-methylerythritol at night in spring was probably due to the enhanced emission of isoprene from biomass burning, which was

consistent with the diurnal patterns of levoglucosan and 4-methyl-5-nitrocatechol in spring." (Section 3.2.2).

"12. Line 303-305 and also Figure 5, one may see that in winter SOC contributed to about 50%, 60% and 65% of PM1-associated, PM2.5-associated and PM10-assocciated WSOC, respectively. In other words, SOC was more enriched in 2.5-10 um size of coarse particles. Why?"

Response: The increasing percentage of SOC with particle size in winter may be due to the fact that hygroscopic growth and the subsequent heterogeneous reactions on the surface of pre-existing particles played a dominant role on SOC formation during the haze episode in winter. Thus organic carbon in larger particles was more in the secondary form. We have added a separate section (Section 3.3.3 Implications for secondary formation based on SOC/OC ratio) to better elucidate the formation mechanism of SOC in PM1, PM2.5 and PM10 during haze episodes in winter and spring respectively.